# Chlorophylls and Polyphenols: Non-Enzymatic Regulation of the Production and Removal of Reactive Oxygen Species, as a Way of Regulating Abiotic Stress in Plants

**DOI:** 10.3390/ijms26189039

**Published:** 2025-09-17

**Authors:** Bogdan Radomir Nikolić, Sanja Đurović, Boris Pisinov, Vladan Jovanović, Danijela Šikuljak

**Affiliations:** 1Institute for Plant Protection and Environment, Teodora Drajzera Str., No. 9, 11040 Belgrade, Serbia; stojakovicsm@yahoo.com (S.Đ.); boriss752002@yahoo.com (B.P.); dulekaca@yahoo.com (D.Š.); 2Institute for Pesticides and Environmental Protection, Banatska Str., No. 31b, 11080 Belgrade, Serbia; vladanjo11@gmail.com

**Keywords:** abiotic stress, prooxidative and antioxidative potential, chlorophylls, polyphenols

## Abstract

Chlorophylls, which are associated with carotenoids and photosynthetic protein complexes, acquire optical properties that enable the absorption of sunlight, necessary for the synthesis of energy and redox equivalents, necessary for photosynthetic absorption of CO_2_ and the production of oxygen as an intermediate product. These processes are important for plants, but also for the biosphere. In stressful situations, when photosynthesis is limited, the production of reactive oxygen and other species increases, and the activation of various protective systems is necessary to remove the aforementioned reactive species or reduce the excessive reduction in photosynthetic electron transport, as the cause of the production of the reactive species. A review of studies where the content and physiological state of chlorophyll are monitored, using destructive and non-destructive methods, such as various optical methods for monitoring its content and physiological activity, is given. Polyphenolic compounds belong to non-enzymatic systems for quenching the reactive species. In addition to their presence in monomeric and oligomeric forms of polyphenols, polymerization of this type of compound can occur. In addition to having a protective effect on the plants that synthesize them, polyphenolic compounds also have a beneficial effect on the health of animals and humans who consume them from plants.

## 1. Introduction

It is known that during the process of photosynthesis in plants, oxygen and other reactive species are produced, which induce oxidative stress. This is because the production of the end products of the so-called “light phase” of photosynthesis (ATP, NDPH_2_) that are required during CO_2_ fixation (directly or indirectly during the so-called C_3_ (Calvin cycle) i.e., C_4_ photosynthesis), also leading to the production of molecular oxygen (O_2_) as a by-product. During optimal photosynthesis of plants, this oxygen is released from the plants, but in a state of stress, when the efficiency of photosynthesis decreases, this molecular oxygen can be a problem. Oxygen can reduce the efficiency of photosynthesis in two ways, through cyclic and pseudocyclic (or Mehler reaction) electron transport around PS_1_, as well as through photorespiration. In the first process, electrons either cycle around PS_1_ or end up at O_2_, as the final electron acceptor, producing reactive oxygen species or phytotoxic H_2_O_2_, while in the second case, RuBP binds not to CO_2_, but to molecular oxygen, which also reduces the efficiency of photosynthesis. The consequence of the production of oxygen during photosynthesis is that although plants ultimately produce sugars and increase organic matter (needed not only for the growth of the plants themselves, but also for the entire biosphere) through this biosynthetic pathway from intermediates of the Calvin cycle and CO_2_ from the air, and although this oxygen as a by-product of photosynthesis allows for more efficient respiration, one of the “dark sides” of this process is the production of reactive oxygen and other species, which can cause photooxidative processes and ultimately lead to the inactivation and degradation of plant cellular structures.

Therefore, during the evolution of plants, mechanisms have evolved to remove the aforementioned oxygen radicals and other reactive molecular species. These mechanisms of protection against oxygen and other reactive species (such as chlorophyll a radical: Chl a*) are divided into enzymatic and non-enzymatic protective systems. One type of molecule that participates in non-enzymatic defense systems is polyphenolic compounds, as quenchers of the aforementioned oxygen radicals and other reactive molecular species. Polyphenols act protectively on plant structures in terms of their protection against oxygen and other reactive species in a number of different ways, which will be discussed in this study.

## 2. Observations Regarding the Applied Research Methodology

The concentration (content) and physiological activity of chlorophyll can be monitored in plants in several ways, using destructive and non-destructive methods. Destructive methods destroy photosynthetic tissue, thereby releasing photosynthetic pigments, primarily chlorophylls a and b (Chl a and b) and different carotenoids, and then their content in the photosynthetic tissue extract (most often in various organic solvents) is determined spectrophotometrically based on the optical properties (absorbance) of photosynthetic pigment molecules [1]. Photosynthetic protein complexes containing Chl a and b, as well as carotenoids, can also be extracted and quantified in a destructive manner [2].

It has been suggested relatively recently that chlorophyll content can be determined non-destructively, whereby the chlorophyll content in intact photosynthetic plant tissue can be determined based on the same optical properties [3,4]. It should be noted that it is still necessary to periodically check these non-destructive methods by using destructive methods applied to the same photosynthetic tissue. Although this non-destructive optical method is considered to be less precise in determining chlorophyll content, it is still often used because it is possible to evaluate a large number of samples in a short period of time, with a relatively minimal consumption of organic solvents [5,6,7,8,9].

It is also worth mentioning other types of optical non-destructive assessment of the content of photosynthetic pigments (primarily chlorophyll) in the leaf cover of entire fields of cultivated or phytocenoses of native plants not in the “classical” visible spectrum, but in the far-red (FR) or near-infrared (NIR) part of the solar spectrum, which is the part of the spectrum emitted by different molecules (chlorophylls, and also many polyphenols) of photosynthetically active plant communities, cultivated, but also native [10,11,12,13,14,15,16,17]. Based on the ratio of the maximum emission of leaves of native and cultivated plants in different parts of the FR and NIR parts of the solar spectrum, different so-called spectral indices have been defined.

Another way to detect the physiological activity of chlorophyll molecules is to use various methods of measurements of chlorophyll fluorescence (but also some polyphenolic compounds; see below), as a way to assess the efficiency of photosynthetic processes in plants [18,19,20,21]. Some of the aforementioned Chl a fluorescence methods were also tested in our research [22,23,24].

It has been mentioned that during plant photosynthesis, especially under stress conditions [25,26], the efficiency of photosynthesis decreases, so that photosynthetically produced oxygen can receive an electron from photosynthetic electron transport [27,28,29,30], thereby producing reactive oxygen and other species, which lead to damage to plant cell structures. The production of reactive oxygen and other species can also be measured by various methods [31,32,33].

In addition to several other enzymatic and non-enzymatic methods for scavenging reactive oxygen and other species [34,35], polyphenol molecules, either in dissolved form in the cytosol of plant cells or as components of the cell wall [36], also serve as quenchers of such reactive species. Various methods for extracting and measuring the content of these compounds in plant tissue samples have been tested [37,38,39,40].

## 3. Photosynthesis, Chlorophylls, and Other Photosynthetic Pigments and Their Interaction with Oxygen Metabolism

Photosynthetic electron transport, occurring on the inner membranes of chloroplasts, produces the end products of the so-called "light phase" of photosynthesis (ATP, NADPH_2_), required during CO_2_ fixation (directly or indirectly during the so-called C_3_ (Calvin cycle) i.e., C_4_ photosynthesis), but also the molecular oxygen (O_2_), by-product.

During optimal photosynthesis of plants [41] that are not under stress, this oxygen is released from the plants, but in a state of stress, when the efficiency of photosynthesis decreases, this molecular oxygen can be a problem. Oxygen can reduce the efficiency of photosynthesis in two ways, through cyclic and pseudocyclic (or Mehler reaction) electron transport around PS_1_ [42,43,44,45], as well as through photorespiration [46,47]. In the first process, electrons either cycle around PS_1_ or end up at O_2_ as the final electron acceptor, producing reactive oxygen species or phytotoxic H_2_O_2_, while in the second case, RuBP binds not to CO_2_, but to molecular oxygen, which also reduces the efficiency of photosynthesis. The consequence of oxygen production during photosynthesis is that although plants ultimately produce sugars and increase organic matter (needed not only for the growth of the plants themselves, but also of the entire biosphere) through this biosynthetic pathway from intermediate products of the Calvin cycle and CO_2_ from the air, and although this oxygen as a by-product of photosynthesis enables more efficient respiration, one of the “dark sides” of this process is the production of reactive oxygen and other species, which in turn attenuates the activation of protective processes, which remove these reactive species [30,34,35] or reduce the excessive reduction in photosynthetic electron transport [20,24,25,48,49], thereby reducing the production of reactive oxygen and other species.

Figure 1 shows computer-generated (based on the original graphical record of Chla fluorescence) unpublished examples of the modulated chlorophyll fluorescence method, which clearly show that the limitation of photosynthesis in wheat leaves under drought stress significantly affects the change in fluorescence parameters, as one of the measures of photosynthetic activity. The aforementioned record is equivalent to the records cited in previous studies [18,19,21]. The fluorescence yields in the record shown in Figure 1 can be related to the fully oxidized RC PS_2_ of darkened leaves (F_0_), then to the fluorescence yield in a darkened leaf exposed to a light flash (complete reduction in RC PS_2_ leaves: Fm), then to the fluorescence yield of RC PS_2_ of illuminated leaves (Fs) in equilibrium photosynthesis, as well as the fluorescence yields of fully oxidized (F_0_^,^) and fully reduced (Fm^,^) RC PS_2_ in illuminated leaves, defined as in references [19,21], whereby various Chla fluorescence parameters are calculated from these fluorescence yields in the oxidized and reduced RC PS_2_ of darkened and illuminated leaves, and thus defining the state of the photosynthetic apparatus in vivo and in situ, determined by the so-called modulated Chla fluorescence (PAM) method. In addition to the PAM method of measuring Chla fluorescence, there are also different types of so-called methods of unmodulated Chla fluorescence [20,26,50,51,52,53,54,55].

The limitation of photosynthesis due to stress (here, herbicide stress) affects the content of photosynthetic pigments and photosynthetic pigment–protein complexes, determined by destructive methods, as can be seen in Table 1 and Table 2. The applied herbicide diquat induces the production of reactive oxygen species [28], which disrupts cellular metabolism, but also causes pheophytinization, i.e., the loss of Mg ions from chlorophyll molecules, which contributes to the formation of radical species based on chlorophyll molecules [56,57,58]. All this leads to accelerated degradation, primarily of Chl a, which depends on the light environment (Table 1). In addition, significant changes in photosynthetic membrane proteins under the action of diquat have been observed, mediated primarily by far-red light (Table 2), which is primarily absorbed through PS_1_ [2].

In recent years, instead of the destructive method of extracting photosynthetic pigments by destroying plant tissue, with the help of organic solvents, various “chlorophyll meters” have been used much more, which, based on the absorbance of leaves in a certain part of the spectrum (most often the red part of the sun spectrum, where the maxima of light absorption by chlorophyll are located), provide an estimate of the chlorophyll content in plant leaves in vivo, either in relative (so-called SPAD meters) or absolute values of chlorophyll content, with the destructive method being used only to determine reference values, against which the aforementioned devices are calibrated [6,9,16].

This method makes it possible to determine in vivo the chlorophyll content in the leaves of various plants, whether it is to determine the resistance of weeds to various herbicides (Table 3 and Table 4) [3,4], but also for determining chlorophyll content, as a measure of the health of cultivated [5,13] plants. The method of calibrating “chlorophyll meters” is presented in several papers [7,8]. It is also important to mention the in vivo remote determination of chlorophyll content, as well as other photosynthetic and some protective pigments, based on field imaging of leaf cover of cultivated plants or native plants in phytocenoses [10,11,12,13,14,15,59,60], either in plants under stress or normal physiological conditions.

Unlike measuring chlorophyll absorbance or its emission in the far-red (FR) or near-infrared (NIR) part of the solar spectrum, determining the fluorescence of Chl a [25,48,49], as well as some polyphenols [20,26], reveals the functional characteristics of the photosynthetic apparatus and the chlorophyll molecules within it. Modern methods of measuring Chla fluorescence are based on the so-called modulation principle, where the plant is simultaneously illuminated with continuous, s.c. actinic light, occasionally interrupted by strong light flashes, which lead to saturation of the so-called reaction center (RC) PS_2_, which allows us to differentiate the so-called photochemical from the so-called non-photochemical quenching of fluorescence (Figure 1), and which has also been described in Refs. [18,19,21].

The first process reflects the productive side of the “light phase” of photosynthesis and shows the approximate equivalent of ATP production, while the second process is a measure of protective processes in the “light phase” of photosynthesis, which prevent excessive reduction in photosynthetic complexes, which would cause oxidative shock. Figure 2 [22] shows the sigmoid relationship between chlorophyll content and RC PS_2_ quantum yield (parameter Fv/Fm), which is very similar to the conclusions of the classic study by Björkman and Demmig [50].

We later (Figure 3) tested the regression dependence of the non-photochemical fluorescence quenching parameter (NPQ) and the photochemical efficiency parameter (RFD 730) on the Gibbs free energy (ΔG_0_) and found a relatively weak regression dependence of these parameters of Chl a fluorescence of maize leaves on the total energetics of that plant in the late vegetative stage of development, at a physiologically low temperature for that species [24].

In addition to the aforementioned modulated Chl a fluorescence, which has been prevalent in recent photosynthesis research, we should also mention the so-called unmodulated fluorescence (so-called OJIP test), practiced by a number of well-known researchers [50,51,52,53,54,55]. In addition, it is worth mentioning fluorescence imaging [20,61,62], where leaf photosynthesis is analyzed in great detail, whether it is cultivated [20,63] or native plants [26,62,64]. It should also be mentioned that chlorophyll fluorescence in various forms can be combined with other methods for detecting the physiological activity of photosynthesis and leaves, which we will not elaborate on in this study [65,66,67,68,69,70,71,72].

Although the production of reactive oxygen species and oxidative stress is mainly related to photosynthesis [27,28], the removal of the reactive species thus produced is carried out enzymatically [34] and non-enzymatically [30,35] or reduces the excessive reduction in photosynthetic electron transport [20,25,26,48,49], which ultimately leads to the production of reactive oxygen species. It is possible that other physiological processes also produce the aforementioned reactive species and oxidative stress, such as during respiration [73] or during cell division and elongation, due to the action of phytohormones [74,75,76]. The aforementioned examples indicate the degradative [29,77], but also productive function of oxygen and other reactive species, usually by activating various physiological processes with other signaling systems [78]. Various types of stress lead to the interaction or production of reactive oxygen and other species, such as in the presence of various xenobiotics [56,79,80], drought and other osmotic stresses [81,82,83], and temperature [83,84] stresses. In short, the relationship between various stresses and the production of oxygen and other reactive species is never a simple process, but involves the activation of many signaling and gene pathways [85,86,87]. Therefore, we present here this small example (Figure 4), as well as some other findings, which illustrate the interrelationship between the production of reactive oxygen and other species, hormonal, as well as other signaling systems, and their impact on plant growth and physiology [31,33,50,88].

## 4. Polyphenols, Their Synthesis, Measurements, and Antioxidative Activity

As mentioned earlier [38], polyphenolic compounds belong to non-enzymatic systems for quenching reactive oxygen and other species.

They are the second most abundant group of organic compounds in plants, and about 10,000 of these compounds had been identified by 2023 [89]. Polyphenolic compounds belong, together with alpha-tocopherol and beta-carotene, to the group of non-enzymatic antioxidants. It is believed that the antioxidant activity of polyphenolic compounds is primarily the result of their ability to be hydrogen donors, after which less reactive phenoxyl radicals are formed. The relatively high stability of phenoxyl radicals is explained by electron delocalization with the existence of multiple resonance forms.

It is thought that part of the antioxidant potential of many types of plants can be attributed to polyphenolic compounds. Polyphenolic compounds exhibit antioxidant activity in biological systems in several ways, namely the following:By handing over H-atoms, which directly bind (“trapping” and/or “quenching”) free oxygen or nitrogen radical species;By chelation of prooxidative metal ions (Fe2+, Cu2+, Zn2+ and Mn2+);By activating antioxidant enzymes;By the inhibition of pro-oxidative enzymes (lipoxygenase, NAD(P)H oxidase, xanthine oxidase, cytochrome P-450 enzymes) [38,90,91,92,93,94].

The synthesis of polyphenolic and other aromatic compounds is an integral part of the major biosynthetic shikimate pathway, as part of primary metabolism [95], which synthesizes the simplest polyphenols, i.e., polyphenolic acids [94,96,97]. as well as the phenylpropanoid biosynthetic pathway, which is part of secondary metabolism [96], and a whole range of different groups of polyphenolic compounds are then synthesized through this biosynthetic pathway [93,96].

In addition to being present in photosynthetic tissue [37], polyphenolic compounds are abundant in the flowers and fruits of plants [90,97]. Also, in addition to their presence in the form of monomeric and oligomeric forms of polyphenols, polymerization of this type of compound can occur, resulting in lignin, one of the most widespread compounds in the living world [91], with a significant role in plant response to various types of stress. It is noteworthy that polyphenolic compounds not only have a protective effect in plants that synthesize them [89,92,93,94,96,98,99], but they also have a beneficial effect on the health of animals and humans who consume them with plant foods [38].

In addition to the natural variability of polyphenols of different classes in native plants [60], the selection of different cultivars based on their content of polyphenolic compounds is also of interest [100]. Besides selecting different cultivars for polyphenol content, it is also important to find appropriate agrotechnical measures that increase the polyphenol contents in edible plant parts, which is a kind of polyphenolic biofortification of plant fruits [38,101]. Therefore, it is important to develop more efficient methods for the detection of various classes of polyphenols [39,40].

The functions of these different classes of polyphenolic compounds are diverse, and they mainly participate in increasing the resistance of plants to various abiotic and biotic stresses, either directly by immobilizing, i.e., preventing the effects of these stresses by creating a kind of barrier (chelation of heavy metals or a kind of “shield” against intense visible or UV light) from their negative effect on cellular processes in plants or by acting as quenchers of reactive, primarily oxygen species, thereby increasing the resistance of plants to stresses [89,91,93,94,99]. Furthermore, since lignins, as a more complex class of polyphenols, participate in remodeling of the secondary cell wall during these processes, it is clear that polyphenols also participate in some developmental processes in plants [93,95].

The antioxidant properties of polyphenols have been studied in cultivated [90,97,100], as well as native [92,98] plants, and it has been observed that different classes of polyphenols perform different protective and developmental functions in plants, depending on the class of polyphenolic compounds [89,91,93,94,99].

The antioxidant capacity of polyphenols was tested experimentally in two ways in the extract of soybean seed meal from soybean plants treated with different forms of plant extract-based fertilizers (rich in polyphenolic substances) (a) by the Briggs–Rauscher titrimetric method [38,101] and (b) by determining the content of total polyphenols and antioxidant activity of the said soybean seed meal extract (DPPH and FRAP tests) [38,101].

In addition to the analysis of the content of total polyphenols and their antioxidant activity, the content of some phenolic acids (gallic (gall), chlorogenic (chl), caffeic (caff), p-coumaric (coum), ferulic (fer) and trans-cinnamic (cinn) acids; [38]) was also determined in the soybean seed extract obtained from the treated plants. A PCA analysis of the statistical effect of treatment with different products based on plant extracts (rich in polyphenolic substances) on soybean plants and their influence on the content of the above-mentioned polyphenolic acids was also performed [38,101], with very different effects of the mentioned preparations on the content of the tested polyphenolic acids [101]. In short, it is possible to carry out biofortification, i.e., enrich soybean seeds with various polyphenolic compounds and thus increase their antioxidant capacity, by applying fertilizers based on various plant extracts.

## 5. Further Research Directions

Due to the optical properties of Chl a and Chl b, their association with various forms of carotenoids (and other plant pigments) and photosynthetic protein complexes, photosynthesis is an equilibrium process, dependent on other phytophysiological processes [102], but it is crucial in plant biology [103], and also for the entire biosphere [104]. Due to the key role of photosynthesis and associated production of oxygen as equilibrium processes in the overall metabolism of plants, in stressful situations, when photosynthesis is limited, oxygen and other reactive species [27,28] can lead to the damage of cellular structures in plants. Therefore, a number of enzymatic and non-enzymatic systems [34,35,105] are necessary to remove those reactive species or to reduce the excessive reduction in photosynthetic electron transport, which leads to a decrease in the production of oxygen and other reactive species. Therefore, it is necessary to monitor the content and physiological state of chlorophyll in plants, using destructive and non-destructive methods, the most important of which are various forms of optical methods for monitoring the state(s) of chlorophyll(s) in plants. These methods are continuously being improved and tested in parallel with other, molecular methods, whereby the physiological activity of chlorophyll in plants with mutated genes for photosynthetic proteins (e.g., [106]) is monitored. Imaging methods are of particular importance, i.e., for the detection of physiological activity in the whole leaf canopy of cultivated plants in the field or phytocenoses of native plants in nature [10,11,12,13,14,15,60,107]. So far, in vivo methods (EPR, etc.) for detecting the production of oxygen and other reactive species are limited to small plants (seedlings of cultivated plants or small plants such as *Arabidopsis*), but there is no reason why we should not use genetically modified plants, e.g., plants transformed with gene(s) for the ”reporter protein(s)” [108,109] that are activated by the appearance of oxygen and other reactive species and signal these phenomena in situ, at the whole plant level, thus enabling the better study of the physiology of production and quenching of oxygen and other reactive species.

Polyphenolic compounds belong to non-enzymatic systems for quenching reactive oxygen and other species. Polyphenolic compounds are present in photosynthetic tissue, but also in flowers and fruits of plants. In addition to monomeric and oligomeric forms of polyphenols, there are also polymeric forms of these compounds, the most complex of which is lignin(s), the second most widespread compounds in plants, with a significant role in plant response to various types of stress. The method of detecting antioxidant activity of polyphenols is most often indirect, either by monitoring the reduction in the production of oxygen and other reactive species or by monitoring the reduction in degradative effects of oxygen and other reactive species on cellular structures of plants. We believe that the use of genetically modified plants with altered contents of various polyphenolic fractions, and subsequent monitoring of antioxidative activities in the tissue of these plants, is a promising path [89,91,93,94,99].

## 6. Conclusions

The problem of photosynthetic oxygen production and the subsequent production of oxygen and other reactive species when photosynthesis is limited under stress conditions is one of the most important problems in plant sciences. Developing methods for detecting and measuring these reactive species, as well as the photosynthesis that produces them, is an important methodological challenge. It is also important to directly or indirectly monitor the various non-enzymatic and enzymatic systems for the removal of these reactive oxygen and other species, including the functions of polyphenols in plants.

## Figures and Tables

**Figure 1 ijms-26-09039-f001:**
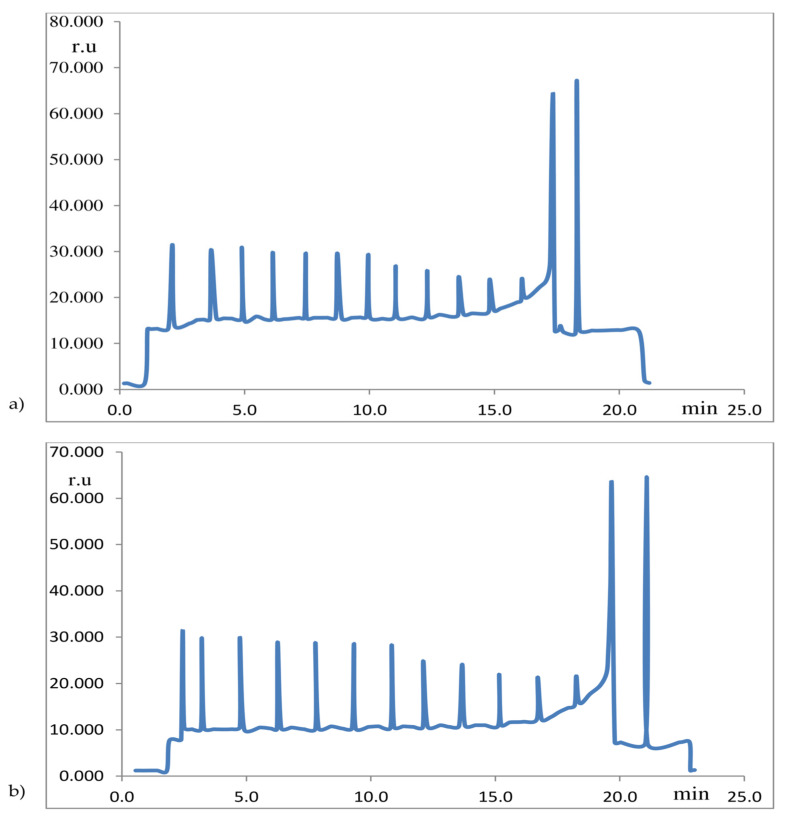
Computer-generated (based on the original graphical record of Chla fluorescence) unpublished examples of the modulated chlorophyll fluorescence method. Modulated Chla fluorescence curves in *T. aestivum* (L. cv. Kg-56) leaves, induced by repeated application of saturating light pulses, already illuminated by continuous actinic light, during drought stress. Abscissa: time (min.); Ordinate: fluorescence intensity (relative unit: r.u.). (Due to the limitations of the computer program for generating Chla fluorescence records, the time axis (abscissa) is directed opposite (from left to right) to the dynamics of the change in Chla fluorescence yield, which occurs from right to left). (**a**) Computer-generated record of modulated Chla fluorescence of wheat leaves after 1 day of drought; (**b**) Computer-generated record of modulated Chla fluorescence of wheat leaves after 4 days of drought. Examples of plant stress were detected by the modulated chlorophyll fluorescence method: altered Chla fluorescence yield was observed in leaves of wheat plants exposed to 4 days of drought (**b**), compared to leaves of the same plants in the initial stage of drought stress (**a**). A record of modulated Chla fluorescence equivalent to that recorded in references [18,19,21].

**Figure 2 ijms-26-09039-f002:**
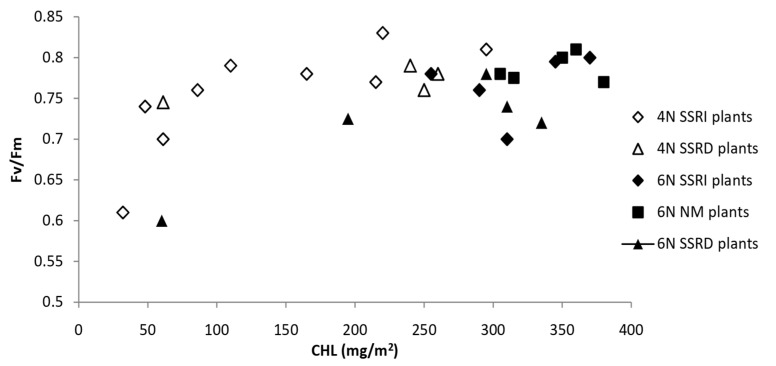
Relationship of total chlorophyll content (mg m^−2^) and quantum yield of RC PS_2_ (Fv/Fm) in young fully developed leaves of four- or six-week-old *Z. mays* plants. Created equivalent to reference No.: [22]. N—week; SSRI—plants in which the source–sink ratio increases; SSRD—plants in which the source–sink ratio decreases. NM: plants not exposed to SSRI, i.e. SSRD treatments.

**Figure 3 ijms-26-09039-f003:**
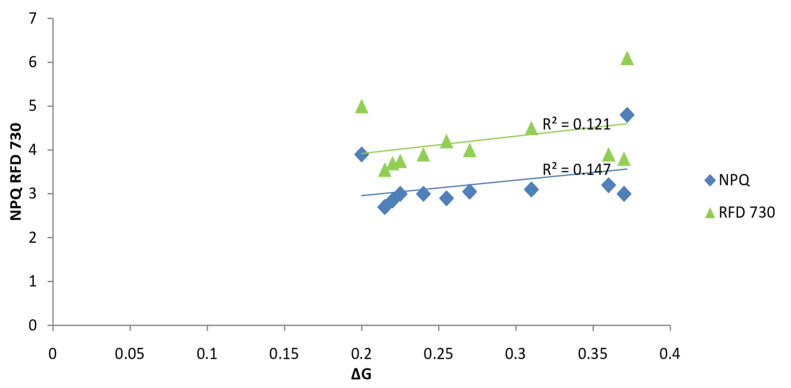
Regression between thermodynamic parameter ΔG 105° and photosynthetic parameters NPQ and RFD 730. Created equivalent to reference No: [24].

**Figure 4 ijms-26-09039-f004:**
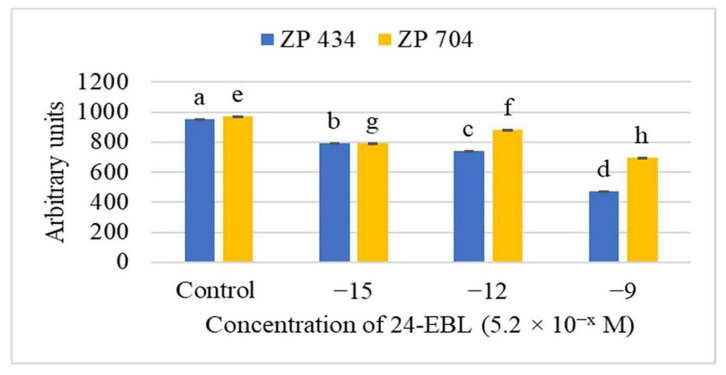
Effect of different concentrations of 24-EBL on total redox status of whole seedlings of ZP 704 and ZP 434 maize hybrids, determined using the EPR spectroscopy method. Created equivalent to reference No: [32]. The results are given in arbitrary units (r.u.) of the EPR signal, double integral values per sample mass. The axis values are displayed in ten thousands. Values indicated by the same letter are not statistically different (*p* < 0.05).

**Table 1 ijms-26-09039-t001:** Effects of diquat in the dark or under “white light” (WL) or far-red radiation (FR) during different time intervals (0–48 h) on ratios of contents of chlorophylls (Chl a and b) and total carotenoids (Carr) in primary leaves of soybean and first leaf of maize. n.m.: not measured. Table created equivalent to reference [2].

Crop	Soybean						Maize									
Parameter	Chl a/b			Chl a/Carr			Chl a/b					Chl a/Carr				
Treatment/treatment time (h)	0	24	48	0	24	48	5	12	18	36	48	5	12	18	36	48
WL	2.1	2.3	2.4	5.3	5.6	4.6	2.6	2.5	2.3	2.5	n.m	4.9	4.8	4.9	5.5	15.3
WL + diquat	2.5	2.0	1.5	4.4	6.6	8.9	2.6	2.6	2.3	1.5	0.8	4.7	5.4	6.1	9.8	35.8
FR	2.2	2.1	1.7	3.4	3.4	3.9	2.6	2.4	2.4	2.1	2.1	4.9	4.6	4.9	4.6	4.9
FR + diquat	2.1	2.1	1.5	3.5	4.4	6.9	2.6	2.4	1.5	1.9	0.9	5.2	4.6	15.8	6.8	11.6
Dark	2.1	2.3	2.6	5.3	4.1	4.2	2.8	2.7	2.5	1.6	1.2	5.0	4.7	5.2	5.7	6.0
Dark + diquat	2.3	2.4	2.1	4.1	4.1	5.1	2.7	2.6	2.4	1.4	1.0	4.9	3.6	5.7	13.8	10.7
LSD_0_._05_	0.07			0.09			0.24					2.13				

**Table 2 ijms-26-09039-t002:** Effects of diquat in the dark or under “white light” (WL) or far-red radiation (FR) during different time intervals (0–24 h) on polypeptides of photosynthetic reaction centers (RC PS_1_ and RC PS_2_) and light-harvesting complexes (LHC) of photosystem PS_1_ and PS_2_ of soybean and maize chloroplast thylakoids. Table created equivalent to reference [2].

Crop	Soybean										Maize									
Parameter	PS_1_		PS_2_		LHC		PS_2_/PS_1_		LHC/PS_2_		PS_1_		PS_2_		LHC		PS_2_/PS_1_		LHC/PS_2_	
Treatment/time (h)	0 h→24 h		5 h→24 h		5 h→24 h		5 h→24 h		5 h→24 h		0 h→24 h		5 h→24 h		5 h→24 h		5 h→24 h		5 h→24 h	
WL	7.1	3.0	23.2	38.6	35.4	34.3	3.3	1.8	1.7	1.0	6.9	5.2	22.2	33.8	36.7	29.8	3.2	6.5	1.7	0.8
WL + diquat	4.0	2.6	19.1	25.4	40.7	43.7	4.8	9.8	2.1	1.7	1.9	4.0	24.7	33.6	42.6	25.6	13.0	8.4	1.7	0.8
FR	5.3	2.6	24.6	23.8	23.9	43.7	4.6	9.2	1.0	1.8	2.9	5.8	28.4	25.7	37.1	27.9	9.8	4.4	1.3	1.1
FR + diquat	3.5	2.0	12.9	15.0	50.7	34.3	3.7	7.5	3.9	1.4	1.2	3.6	21.2	23.4	45.9	23.4	17.7	6.5	2.2	1.0
Dark	3.5	3.1	19.9	22.0	39.0	46.3	5.5	8.7	2.0	1.7	1.5	4.0	25.0	26.3	35.8	30.9	16.7	10.0	1.4	1.2
Dark + diquat	4.9	1.8	34.9	30.0	34.9	37.0	4.7	16.7	1.5	1.2	n.m.	2.6	45.0	32.0	36.3	25.1	45.0	12.3	0.8	0.8
LSD_0_._05_	0.8		10.5		9.3		1.5		1.2		1.1		5.1		8.5		7.6		0.4	

n.m.: not measured. The content of the LHC photosynthetic complex is expressed as % of the sum of proteins with a mass of 25–30 kDa, i.e., total proteins. The content of the PS_1_ photosynthetic complex protein is expressed as % of the sum of all proteins with a mass of 66 and 110 kDa, i.e., total proteins. The content of the PS_2_ photosynthetic complex protein is expressed as % of the sum of all proteins with a mass of 45–50 kDa, i.e., total proteins.

**Table 3 ijms-26-09039-t003:** Total amount of chlorophyll (r.u.) determined by SPAD meter readings. Table created equivalent to reference [3].

Weed Population	Control Plants		5 Days After Treatment (8 kg ha^−1^ Atrazine)			
	Average Amounts		Average Amounts		LSD Test	
	R	S	R	S	C:R	C:S
*Abutilon teophrasti*	24.58	25.27	22.42	23.45	ns	**
*Amaranthus retroflexus*	18.45	22.17	22.80	21.86	ns	**
*Chenopodium album*	36.04	36.38	34.89	30.40	**	**

*p* < 0.01 **, ns—nonsignificant differences, K—control, R—resistant to herbicides weed population, S—susceptible to herbicides weed population.

**Table 4 ijms-26-09039-t004:** Total amount of chlorophyll (r.u.) determined by SPAD meter readings, *Lolium* sp. and *Conyza* sp. populations. Table created equivalent to reference no: [4].

Weed Population	Control Plants		6 Days After Treatment (1 kg a.i.ha^−1^ Glyphosate)			
	Average Amounts		Average Amounts		LSD Test	
	R	S	R	S	C:R	C:S
*Lolium rigidum*	33.43	38.33	37.06	34.69	ns	**
*Conyza canadensis*	-	48.46	-	50.81	-	Ns
*Conyza bonariensis*	-	39.17	-	43.06	-	*

*p* < 0.01 **, *p* < 0.05 *, ns—nonsignificant differences, K—control, R—resistant to herbicides weed population, S—susceptible to herbicides weed population.

## Data Availability

The data presented in this study are available on request from the corresponding author.

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
