# Peer review of "Chlorophylls and Polyphenols: Non-Enzymatic Regulation of the Production and Removal of Reactive Oxygen Species, as a Way of Regulating Abiotic Stress in Plants"

_ijms, 2025, doi:10.3390/ijms26189039_

Round 1

Reviewer 1 Report

Comments and Suggestions for Authors

Dear Prof Dr. Ms. Aurora Guo

Editor

IJMS

Please find enclosed my revision of the Manuscript ID: ijms-3762590

Manuscript title: Pro-oxidative potential of chlorophylls and antioxidant potential of
polyphenols: Non-enzymatic regulation of the production and removal of
reactive oxygen species, as a way of regulating abiotic stress in plants

The present manuscript assessed the role of polyphenolic compounds in quenching reactive oxygen species in plants. The manuscript needs minor revision through

  • Introduce more information about the specific roles of polyphenolic compounds in nullifying reactive oxygen species injury.
  • The author must use high-resolution figures
  • The role of phenolic compounds in boosting chlorophyll levels and their relation with oxidative burst
  • Rewrite the conclusion section and focus on a maximum of 4-5 sentences.

Yours truly

Reviewer 2 Report

Comments and Suggestions for Authors

In the manuscript the authors attempted to review the ways production of reactive oxygen species during photosynthesis and their detection and the role of polyphenols in for quenching reactive oxygen. The topic is interesting, but in my opinion major changes should be made before the manuscript meets the standards for publication in IJMS. 
1) lines 4-5: the title of the article is incomplete. However, it suggests that various abiotic stresses will be considered regarding the generation and removal of reactive oxygen species. In fact the detailed discussion on this topic is lacking.
2) the inclusion of experimental data in the form of tables (1-4) and figures (2-7) into a review is somewhat confusing. I think these tables and figures should be removed and the text edited accordingly.
3) the section devoted to polyphenols (lines 250-296) is very short and principally is the compilation of experimental data from the authors' previous publications. The title of the manuscript suggests that antioxidant potential of polyphenols will be discussed in more details and in connection with the production of reactive oxygen species during photosynthesis. In my opinion this section either should be expanded or removed altogether so that the review concentrates only on methods of detection of ROS production during photosynthesis.

Round 2

Reviewer 2 Report

Comments and Suggestions for Authors

The authors haven't made any substantial changes in the manuscript, and its contents are still not justifying the title of the manuscript. There is no detailed discussion of generation and removal of reactive oxygen species in response to various abiotic stresses. The new text in lines 291-309 does not provide detailed review of antioxidant potential of polyphenols, merely citing a few well-known facts.

Author Response

Dear editors and reviewer, please, find attachment file with replies on reviewers comments. 

Round 3

Reviewer 2 Report

Comments and Suggestions for Authors

I can see that the authors have added the references in several paragraphs, but haven't made any substantial changes to the text. In my opinion these changes are essential to make the manuscript a comprehensive review of the current state of knowledge in the area delimited by the manuscript's title.

Author Response

Dear Reviewer No. 2, we are attaching a document with a response to your constructive comment.
